# Left Ventricular Mass Reduction by a Low-Sodium Diet in Treated Hypertensive Patients [note 1]

**DOI:** 10.3390/nu12123714

**Published:** 2020-11-30

**Authors:** Natale Musso, Federico Gatto, Federica Nista, Andrea Dotto, Zhongyi Shen, Diego Ferone

**Affiliations:** Centre for Secondary Hypertension, Unit of Clinical Endocrinology, Department of Internal Medicine, University of Genoa Medical School, IRCCS Ospedale Policlinico San Martino, 16132 Genova, Italy; fedgatto@hotmail.it (F.G.); nistafodi@libero.it (F.N.); andreadotto91@gmail.com (A.D.); italiashen@gmail.com (Z.S.); ferone@unige.it (D.F.)

**Keywords:** blood pressure, hypertension, left ventricular hypertrophy, left ventricular mass, low-sodium diet, sodium consumption

## Abstract

Objective: To evaluate the left ventricular mass (LVM) reduction induced by dietary sodium restriction. Patients and Methods: A simple sodium-restricted diet was advised in 138 treated hypertensives. They had to avoid common salt loads, such as cheese and salt-preserved meat, and were switched from regular to salt-free bread. Blood pressure (BP), 24-h urinary sodium (UNaV) and LVM were recorded at baseline, after 2 months. and after 2years. Results: In 76 patients UNaV decreased in the recommended range after 2 months and remained low at 2 years. In 62 patients UNaV levels decreased after 2 months and then increased back to baseline at 2 years. Initially the two groups did not differ in terms of BP (134.3 ± 16.10/80.84 ± 12.23 vs. 134.2 ± 16.67/81.55 ± 11.18 mmHg, mean ± SD), body weight (72.64 ± 15.17 vs. 73.79 ± 12.69 kg), UNaV (161.0 ± 42.22 vs. 158.2 ± 48.66 mEq/24 h), and LVM index (LVMI; 97.09 ± 20.42 vs. 97.31 ± 18.91 g/m^2^). After 2years. they did not differ in terms of BP (125.3 ± 10.69/74.97 ± 7.67 vs. 124.5 ± 9.95/75.21 ± 7.64 mmHg) and body weight (71.14 ± 14.29 vs. 71.50 ± 11.87 kg). Significant differences were seen for UNaV (97.3 ± 23.01 vs. 152.6 ± 49.96 mEq/24 h) and LVMI (86.38 ± 18.17 vs. 103.1 ± 21.06 g/m^2^). Multiple regression analysis: UNaV directly and independently predicted LVMI variations, either as absolute values (R^2^ = 0.369; β = 0.611; *p* < 0.001), or changes from baseline to +2years. (R^2^ = 0.454; β = 0.677; *p* < 0.001). Systolic BP was a weaker predictor of LVMI (R^2^ = 0.369; β = 0.168; *p* = 0.027; R^2^ = 0.454; β = 0.012; *p* = 0.890), whereas diastolic BP was not correlated with LVMI. The prevalence of left ventricular hypertrophy decreased (29/76 to 15/76) in the first group while it increased in the less compliant patients (25/62 to 36/62; Chi^2^
*p* = 0.002). Conclusion: LVM seems linked to sodium consumption in patients already under proper BP control by medications.

## 1. Introduction

Hypertension is the primary modifiable cardiovascular risk factor [1,2]. Left ventricular hypertrophy (LVH) is both a common complication of hypertension and a crucial risk factor for cardiovascular morbidity and mortality [3]. Cardiovascular complications in hypertension can be predicted by an increased left ventricular mass (LVM), conversely the regression of LVH during blood pressure (BP) lowering treatment is associated with a reduction of cardiovascular events [4]. LVH is one of many factors involved in the impact of hypertension on cardiovascular diseases [3,5], and its regression should be considered a relevant goal in the anti-hypertensive management [6,7,8]. In the middle of the last century, the efficacy of a non-pharmacological intervention, such as a very low-sodium diet (the so-called rice-fruit diet was actually extremely low in sodium, 0.25 g a day), raised attention and controversies in the academic world [9,10]. Furthermore, during the eighties, the potential contribution of dietary sodium on LVM hypertrophy has been pointed out and it was found to be independent of BP values [11]. In spite of the well-recognized role of the low-sodium diet in the primary approach in the management of hypertension [1,2], how low should the sodium be in a patient’s diet remains a matter of hard debate [12], since some dissenting scientists have claimed unfavorable effects of the (recommended) low-sodium approach [13]. In this context, the recommended intake of sodium (2.3 g a day, as suggested by the international Guidelines [1,2]) has been questioned on the basis of very large studies [13]. Dissenting researchers claimed a sort of J-shaped relationship between sodium intake and cardiovascular events, showing how the Guidelines’ recommended limit would be harmful for patients and suggesting that actual sodium consumption around the world represents the “optimal” sodium intake. Many mainstream scholars criticized the methodology employed to show this J-curve, namely the single baseline spot urinary sodium measurement. The analytic design of such studies seems flawed by this basic systematic error, able to potentially create relationships that curve [12].

Controversies apart, a restriction of sodium intake (by decreasing salt consumption), can favorably impact hypertensive patients’ health, reducing blood pressure, cardiovascular risk [14], total mortality [15], and drug consumption [16]. The risk of cardiovascular disease shows a linear increase for every gram of sodium consumption [14]. The low-sodium approach not only reduced the risk of cardiovascular disease, but also decreased all-cause mortality [15].

Dietary sodium has been proposed to affect the left ventricular size both in normotensive and hypertensive individuals, and to be the best predictor of LVH in hypertension [17]. A low-sodium approach can favorably impact in this setting, since such a diet was associated with a reduction of LVM matching the drug treatment [18], while patients without LVH did not reduce their left ventricular mass [19]. In hypertensive patients already under proper pharmacologic BP control, a low-sodium approach might add further LVM reduction. Therefore, the main aim of this study is to investigate the long-term effects of sodium restriction on the LVM in patients under active BP lowering treatment.

## 2. Materials and Methods

We evaluated 153 hypertensive patients, already involved in a previous research protocol carried out by our group [16]. Briefly, these patients (part of a larger group) had been previously instructed to restrict dietary sodium and they actually reduced their urinary sodium [16], although compliance of participants on the diet was not examined and no dietary assessment had been conducted. In the former study the whole population had the BP and 24-h urinary sodium excretion (UNaV) measured twice, two months apart, before and after the diet administration. Patients with a reduction in UNaV were considered as responders [16], and were enrolled in the present study, after two years (on average our patients are visited twice a year and the diet is recommended again during each visit). Actually, the low-sodium diet prescribed by our dietitian was based on simple recommendations printed on a single A4 sheet. Patients were advised to avoid salty foods, ice cream, cheese and cured meats, such as bacon, ham, and sausage, as well as other packaged and processed foods. Low-sodium bottled water and salt-free bread were recommended (see ref. [16]). One hundred and forty-nine (149) of them accepted and gave their written informed consent to the research protocol approved by our local Ethical Committee (CE_OSM100598). Patients had already undergone a selection process of eligibility, following predetermined criteria: ineligible patients were those with secondary hypertension, congestive heart failure, atrial fibrillation, diabetes, renal failure, serum electrolyte abnormalities. Dropout patients were all those unable to attain a valid repeated automated BP evaluation, or those unwilling or unable to collect a valid 24-h urine output, as well as the patients with 24-h urine volume below 700 mL. Switching of drug classes during the study period led to drop out the patient from the protocol [16]. Four (4) patients were lost during the follow-up, and seven did not follow the predetermined criteria for the final urinary collection (*n* = 11 did not complete the study).

One hundred and thirty-eight (138) patients (78 females and 60 males) completed the study. In all patients, 24-h UNaV and urinary potassium (UKV) were measured immediately before, or shortly after, the BP measurement. Patients, already trained to collect 24-h urine, received written directions. UNaV and UKV were measured by means of commercial Auto Analyzers (COBAS 8000; Roche Diagnostics, Indianapolis, IN). BP was measured as a repeated measure recording [1,2] by means of Omron HEM 907 BP monitors (Omron, Kyoto, Japan), with an appropriate cuff size, and in accordance with the usual recommendations [20]. With the patient in sitting position for 5 min, her/his BP was measured once by a physician (BP1). It was then measured automatically twice more (BP2 and BP3, with an interval of 3 min) while the patient was alone in the room. BP3 values were employed in the present study. Body height and weight were recorded during the visit. All patients were on pharmacological treatment for hypertension. The oral antihypertensive treatment was measured using the World Health Organization (WHO) Defined Daily Dose (DDD), which is the assumed average maintenance dose needed to reduce the BP to a normal level in patients with mild to moderate hypertension [21].

In our patients, echocardiographic left ventricular mass data were available at the time of the previous study [16]. The echocardiographic measurements (in 2D-guided M-mode) were repeated at the time of the recall, by trained sonographers using Acuson Sequoia C-256 us machine (Acuson Corp., Mountain View, CA, USA), Esaote MyLab Sigma, or Esaote MyLab 30 (Esaote S.p.A., Genova, Italy), following current guidelines [22]. Sonographers themselves were not aware of UNaV results.

Body surface area (BSA) was calculated using the old Du Bois formula [23].

The left ventricular mass was calculated with the Devereux’ formula [22,24], and then indexed by BSA instead of height ^(2.7)^ to attain left ventricular mass index (LVMI), since in our population the prevalence of obesity is low [25], under 9% (Table 1). LVH was considered when LVMI was above 95 g/m^2^ in females and above 115 g/m^2^ in males [22].

Standard statistical methods were performed with commercial software package PRISM (which occasionally calculates only *p* < 0.0x, instead of the exact *p* value—GraphPad Software, San Diego, CA): One-way ANOVA followed by Newman-Keuls (data normally distributed) or Kruskal-Wallis post-test, with Dunn’s Multiple comparison (non-parametric data), Wilcoxon and Mann Whitney tests (non-parametric data), Paired Student’s *t* test (data normally distributed), contingency analysis (Chi^2^—binary variables), and linear regression analysis. Multiple linear regression analysis was performed by use of SPSS software version 20.0 for Windows (SPSS, Chicago, IL, USA). Significance level was set at *p* = 0.05. Values are represented as mean ± SD unless otherwise indicated.

## 3. Results

Our patients (*n* = 138) are part of a group of treated hypertensive patients (*n* = 153 from a cohort of 291 patients), administered with a dietary advice, who showed to comply with sodium reduction in a short-term (2 months) observation [16]. Time 0 and Time 1 refer to the prior study: Time 0 is the baseline before the diet administration, and Time 1 is the evaluation performed after two months. Time 2 refers to the present study, two years after the diet administration. After a two-year period slightly more than a half of patients (55%; *n* = 76 out of 138) showed a stable or lower UNaV (UNaV from 161.0 ± 42.22 at Time 0, to 113.6 ± 32.39 at Time 1, to 97.3 ± 23.01 at Time 2, mEq/24 h), while the remaining 45% (*n* = 62 out of 138) showed an increased UNaV (UNaV from 158.2 ± 48.66 at Time 0, to 106.8 ± 36.32 at Time 1, to 152.6 ± 49.96 at Time 2, mEq/24 h). The first group is classified from now on as Reducer (with apparent long-time compliance), while the second is called Non-Reducer (without apparent long-time compliance: even patients with small UNaV increments, as low as 5–10% were included in this group [26]; actually the lowest increment detected was 6%). The descriptive statistics are reported in Table 1 (whole group) and Table 2 (subgroups).

The links among LVMI, UNaV, systolic BP (SBP), diastolic BP (DBP), DDD, age and body weight, showed a significant level for the relation between LVMI and UNaV at Time 0 (r = 0.411; *p* < 0.001) (Figure 1) and at Time 2 (r = 0.557; *p* < 0.001) (Figure 2). As expected, LVMI and body weight were significantly and directly correlated at Time 0 (r = 0.323; *p* < 0.001), and at Time 2 (r = 0.249; *p* = 0.003). On the other hand LVMI and SBP (r = 0.141 and *p* = 0.099 at Time 0; r = 0.121 and *p* = 0.158 at Time 2), LVMI and DBP (r = 0.135 and *p* = 0.115 at Time 0; r = 0.091 and *p* = 0.287 at Time 2), LVMI and DDD (r = −0.058 and *p* = 0.501 at Time 0; r = 0.018 and *p* = 0.832 at Time 2), and LVMI and age (r = −0.054 and *p* = 0.532 at Time 0; r = 0.002 and *p* = 0.981 at Time 2) were not significantly correlated. Table 2 reports the descriptive statistics of the two subgroups: those patients with a persistently low UNaV (those who seem to follow a reliable low-sodium diet: Reducers) and patients with UNaV levels back to the baseline (Non-Reducers).

The two groups did not show any significant difference in terms of BP (systolic or diastolic), UKV, age, weight, or DDD from Time 0 to Time 2 (1-way ANOVA, Kruskal-Wallis, with Dunn’s Multiple comparison test, *p* > 0.05; even the apparently large difference between DDDs at Time 2 did not reach the significance level). They did not differ even in terms of LVMI at baseline (Time 0); Wilcoxon and Mann–Whitney tests: *p* = 0.567 to *p* = 0.755. A significant difference was found in terms of UNaV at Time 2, with a Δ rank sum (1-way ANOVA, Kruskal-Wallis and Dunn’s post-test) of −168.4 and *p* < 0.05 (Figure 3). Another significant difference was identified in terms of LVMI at the end of the study—Time 2, with a lower value in patients who appeared to follow a long-term diet, with a Δ rank sum (1-way ANOVA, Kruskal-Wallis and Dunn’s post-test) of −69.44 and *p* < 0.05. This was confirmed by Wilcoxon signed rank test, sum of signed ranks −1115, two-tailed *p* < 0.001 and Mann Whitney test, U = 0.1243, two-tailed *p* < 0.001 (Figure 3).

Incidentally, the LVMI decreased significantly in Reducers (from 97.09 ± 20.42 to 86.38 ± 18.17 g/m^2^; Paired Student’s *t* test, *t* = 10.39; *p* < 0.001), while it increased in Non-Reducers (from 97.31 ± 18.91 to 103.10 ± 21.06 g/m^2^; Paired Student’s *t* test, *t* = 4.07, *p* < 0.001).

LVMI and UNaV showed significant relationships also in the two subgroups, first evaluated by use of simple linear regression. Using UNaV as a predictor of LVMI changes, in Reducers we found r = 0.270; *p* = 0.018 and r = 0.369; *p* = 0.001 at the baseline (Time 0) and at the end of the study (Time 2), respectively. Similarly, in Non-Reducers we found r = 0.579; *p* < 0.001 and r = 0.497; *p* < 0.001 at the baseline (Time 0) and at the end of the study (Time 2), respectively.

LVMI and BP relationships in the two groups are complex: while in Non-Reducers we did not find any significant relationship (r = −0.093 systolic and r = 0.022 diastolic; *p* = 0.474 and *p* = 0.865, respectively), in Reducers we found significant relationships only at Time 2 (the last observation) and only for SBP (r = 0.362; *p* = 0.001), with non-significant values for DBP (r = 0.16; *p* = 0.169).

LVH was detected in 29 out of 76 Reducer patients at Time 0, while it decreased to 15 out of 76 two years apart, at Time 2. In Non-Reducer patients the figures are: LVH in 25 out of 62 subjects at Time 0 and in 36 out of 62 at Time 2 (Chi^2^
*df* 19.99, 3; *p* = 0.002) (Appendix A). Simply speaking, the LVH prevalence in our patients decreased by 48% when sodium excretion was maintained low, while it increased by 44% when sodium excretion was not controlled, even if BP (Table 2) and drug consumption (Table 2 and Appendix A) remained comparable.

Multiple regression analysis (see Table 3, for Time 2) showed that LVMI is independently influenced by sodium intake (as appeared from UNaV) more than BP or DDD.

In the whole group, at Time 0, LVMI correlated with UNaV (β = 0.407; *p* < 0.001), independently of BP or DDD. At Time 2, LVMI correlated either with UNaV or SBP (β = 0.611; *p* < 0.001; and β = 0.168; *p* = 0.027, respectively), irrespective of DBP or DDD.

In Reducers, at Time 0, LVMI correlated with UNaV (β = 0.255; *p* = 0.033), independently of BP or DDD. At Time 2 LVMI correlated either with UNaV or SBP (β = 0.345 and β = 0.314; *p* = 0.002 and *p* = 0.005, respectively), irrespective of DBP or DDD.

In Non-Reducers, at Time 0, LVMI correlated with UNaV (β = 0.564; *p* < 0.001), independently of BP or DDD. At Time 2 LVMI correlated with UNaV (β = 0.586; *p* < 0.001), independently of BP or DDD. What has been shown for absolute values seems strengthened by the analysis of the changes over two years, where UNaV is the single, direct, and significant predictor of LVMI change (B = 0.177; β = 0.677; *p* < 0.001) (Table 4 and Figure 4).

Overall, multiple regression analysis confirms that sodium intake, expressed by patients’ UNaV, is the main determinant of LVMI. Indeed, although a statistically significant impact of SBP has been detected, the standardized coefficient of UNaV is about 3.5-fold higher compared to that of SBP (UNaV: β = 0.611, *p* < 0.001; SBP: β = 0.168, *p* = 0.027). Furthermore, looking to the correlation between LVMI and patients’ parameters within the Non-Reducer group, both simple and multiple regression analyses point out UNaV as the only significant predictor of LVMI changes, (Table 3 and Table 4).

Finally, the LVMI modifications from baseline to Time 2 were substantially different in Reducers vs. Non-Reducers. While in the former group a decrease has been shown, in the latter we have observed an increase instead. This behavior seems unrelated with the antihypertensive drugs employed (Appendix A), since the contingency analysis showed non-significant differences among pharmacological treatments (Chi^2^
*df* 0.0007 to 0.7745; *p* = 0.739 to *p* = 1.0).

No sex-specific differences were found between the groups by ANOVA, Kruskal-Wallis followed by Dunn’s multiple comparison test: *p* > 0.05.

## 4. Discussion

Half of our patients who were supposed to follow a low-sodium diet in the long run had benefits in terms of BP control, drug reduction and, above all, LVM reduction and LVH regression.

The effects of possible sodium restriction have been observed in our patients: BP control improved and drug consumption decreased, already in the short term [16]. Surprisingly, these effects are maintained in the long term, even in the patients unable to comply after the initial observations. After two years, indeed, there were neither substantial nor significant differences between the two subgroups of patients in terms of BP and drug consumption that remained unchanged in spite of the apparent compliance or noncompliance with the diet.

The only significant and substantial differences between the two subgroups are represented by LVM and UNaV. The two variables show a highly significant and strict relationship, both in the whole group and in the subgroups.

The LVM has been studied in our patients with the standard 2D-guided M mode. Although other methods are capable of more refined results [27,28], the 2D-guided M mode is a reliable one [22,24]. The increase of the left ventricular mass contributes to the prediction of cardiovascular morbidity and mortality, independent of conventional risk factors [29]. Since the pioneering research of Robert Tarazi [30], the regression of left ventricular mass and the reversal of LVH remain key points in the prevention of cardiovascular events and in the treatment of hypertension [1,2,6]. Whether or not the LVM is more responsive to a drug class or another and how much the LVH regression may be related to the amount of the BP decrease is still a matter of debate [7,31,32,33]. LVM decreases by >10% with angiotensin receptor blockers, by 10% with calcium channel blockers or with ACE inhibitors, by <10% with diuretics or with β-blockers [31,32,33]. Notably, studies have shown the favorable effects of a BP decrease on LVH regression, mostly due to the established antihypertensive treatment [6].

In our patients, there was no association between BP and LVM and this might be due to their chronic drug treatment and the resulting acceptable BP values, already before the present observations. Indeed, it’s worth speculating that the paramount effects of the antihypertensive treatments on LVM had been already occurred before the beginning of the present study.

High sodium intake is the top dietary risk factor for cardiovascular disease [15]; sodium and development of LVH are linked by hormonal, inflammatory and immune mechanisms [15,16]. Some experimental studies on animals investigated the pathophysiology of genes and proteins involved in inflammatory and oxidative processes promoted or upregulated by excess sodium [34,35,36]. The early damage to vascular smooth muscle cells, preceding hypertrophy, seems to involve alterations of the glycocalix [37]. This effect is seen after few days of sodium exposure [37].

The regression of the left ventricular mass, and the reversal of LVH, in severe hypertension under dietary sodium restriction are historical data. Already in the papers by Walther Kempner, the rice-fruit diet, with its very low-sodium content, could normalize both the BP values and the cardiac mass [9,10]. More recently, in hypertensive patients who actually restricted their sodium consumption, a reduction of the left ventricular mass was observed [18,19], even approaching the best pharmacological effect [18]. Conversely, an increase in sodium/potassium ratio was associated with a higher LVMI in pre-hypertensive and hypertensive patients [38,39]. The association between sodium intake and worsening of cardiac mass has been receiving widespread attention [15,40,41,42]. The known linear relations between sodium intake and cardiovascular risk [14] and between UNaV and LVH [15] have even shown a stepwise increase, irrespective of BP values [43].

In our patients, the favorable effect of the dietary sodium reduction on LVM seems marginally related to the BP values, owing to the only partial relationships between these variables. This is not surprising in view of the previous literature [5,19,44,45], where effects of sodium intake on the cardiac size were shown to be independent of BP. Experimental data attribute the sodium-induced LVH to actions mediated by the renin-angiotensin system [46], without an increased sympathetic tone [47]. Furthermore, the restriction of alimentary sodium reduces central BP independently of changes in peripheral BP. It follows that a high central BP, a stronger and independent predictor of cardiovascular morbidity and mortality, can induce a cardiac damage unrelated with peripheral BP [48].

Patients observed in the present study seem to show a long-term effect of the dietary approach in terms of BP control and drug consumption, which persists even in those who apparently ceased the diet. This apparently “long-term protective effect” by an earlier (and somehow limited in time) sodium-restricted diet was initially assessed in the Kempner’s cohort [9], and more recently the multicenter TONE study, showed that the benefits of a dietary intervention do persist long term even after the conclusion [49]. All these evidences may be related with the role of the tissue sodium content [50]. Skin sodium content has strong relationships with the ventricular mass, independent of BP [50]. This could happen possibly because skin sodium content reflects heart sodium content, which in turn may be related with LVH [50].

Our data detected a strong relationship between the LVM values and UNaV (and so with the most reliable measure of sodium intake), more than with BP values recorded at the same time, while no relationship exists with DDD. It may be argued that in treated hypertensive patients the LVM may be subjected more to the sodium intake than to the BP levels when these latter are sufficiently controlled by the drug treatment. The LVM reduction, achieved in the long run by those patients apparently compliant to sodium restriction, may be in a range similar to that achieved by usual antihypertensive drugs [7,8]. However, the amount of drug treatment recorded in our patients does not significantly differ between the groups (Table 2 and Appendix A).

Limitations of the study: We have limited data regarding the sodium consumption of our patients because we could analyze only three 24-h urine collections, while more would have been required [51]. Nevertheless, the relationships of UNaV with the cardiovascular parameters here shown seem suggestive of a fair representation of sodium consumption. The long-term compliance remains a trouble in the non-pharmacological approach to cardiovascular prevention [52,53,54].

Strengths of the study: In hypertensive patients already under proper drug treatment, UNaV is significantly related with the LVM. This relationship seems independent of BP values. Our data support the importance of reducing the dietary sodium in hypertensive patients, even in those under proper BP control.

## 5. Conclusions

LVM decrease (approximately by 11%) under dietary sodium reduction is comparable to what is attained with drug treatment. In hypertensive patients already under active drug treatment and able to follow a dietary sodium restriction, the prevalence of LVH decreases by about half. Without sodium restriction, LVH prevalence increases by a similar extent. In hypertensive patients under chronic drug treatment, the favorable effect of dietary sodium restriction on LVMI seems independent of BP.

## Figures and Tables

**Figure 1 nutrients-12-03714-f001:**
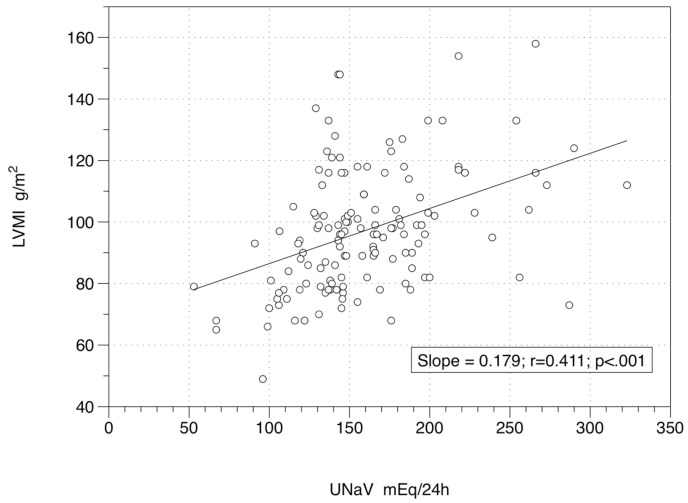
Relationship between left ventricular mass index and 24-h sodium excretion at baseline. Figure 1 Legend: 138 patients under active drug treatment at Time 0, when a low-sodium diet was administered. Equation of the regression line is y = 68.53 + 0.18x. UNaV: 24-h urinary sodium (excretion); LVMI: left ventricular mass index.

**Figure 2 nutrients-12-03714-f002:**
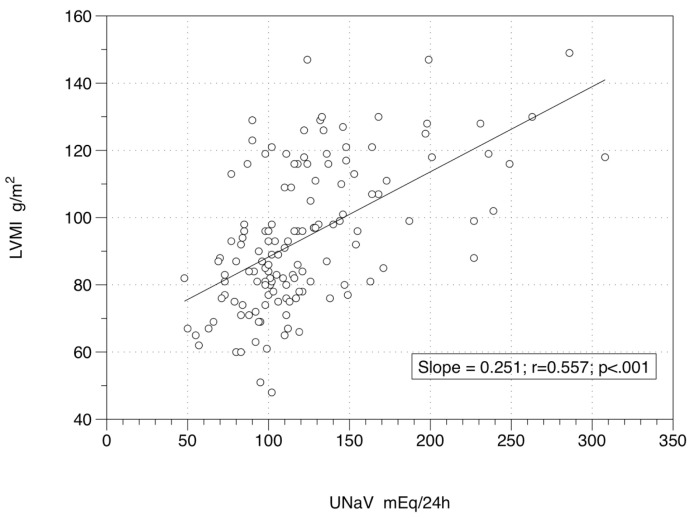
Relationship between left ventricular mass index and 24-h sodium excretion at end of study. Figure 2 Legend: 138 patients under active drug treatment at Time 2, two years after a low-sodium diet was administered. Equation of the regression line is y = 62.97 + 0.25x. UNaV: 24-h urinary sodium (excretion); LVMI: left ventricular mass index.

**Figure 3 nutrients-12-03714-f003:**
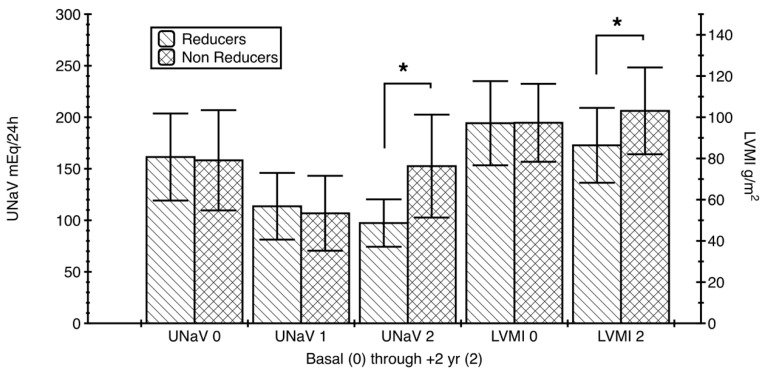
Left ventricular mass index and 24-h sodium excretion modifications in the two groups of patients. Figure 3 Legend: Reducers (*n* = 76) had a significant LVMI decrease (two-tailed Student’s *t* test *p* < 0.001), while Non-Reducers (*n* = 62) had a significant increase (two-tailed Student’s *t* test *p* < 0.001) of their LVMI over the observation period (2 years). All this was paralleled by the changes of the 24-h sodium excretion. The differences between the groups were significant for LVMI and UNaV at Time 2 (1-way ANOVA, Kruskal-Wallis followed by Dunn’s multiple comparison test: both * *p* < 0.05). UNaV: 24-h urinary sodium (excretion); LVMI: left ventricular mass index.

**Figure 4 nutrients-12-03714-f004:**

The graphical representation is reported as an example for simple linear regressions (Δ LVMI vs. Δ UNaV, Δ SBP, and Δ DBP, respectively).

**Table 1 nutrients-12-03714-t001:** Descriptive statistics of the whole patient group.

	Time 0 Mean ± SD	Time 1 Mean ± SD	Time 2 Mean ± SD
Weight kg	73.16 ± 14.07	71.51 ± 13.41 *	71.30 ± 13.21
BMI kg/m^2^	26.19 ± 4.71	25.61 ± 4.39 *	25.50 ± 4.25
SBP mmHg	134.3 ± 16.29	125.1 ± 9.84 §	125.0 ± 10.33
DBP mmHg	81.16 ± 11.73	75.36 ± 8.19 §	75.08 ± 7.63
LVMI g/m^2^	97.19 ± 19.68	-----------------	93.91 ± 21.17 **
UNaVmEq/24 h	159.8 ± 45.08	110.6 ± 34.25 §	122.1 ± 46.52 #
UKV mEq/24 h	61.38 ± 27.72	57.22 ± 22.43	61.25 ± 17.49
DDD	1.772 ± 0.918	1.308 ± 0.731 §	1.343 ± 0.728

Whole Group: Time 0 is the initial observation; Time 1 is +2 months; Time 2 is +2 years; *n* = 138; (Females/Males: 78/60); Body Mass Index ≥ 30, *n* = 12 (8.695%); Age 64.59 ± 12.39 years. (at Time 2). Statistics: ANOVA followed by Newman Keuls post-test: * *p* < 0.05; # *p* < 0.01; § *p* < 0.001 (vs. the respective previous time). Two-tailed Student’s *t* test for paired data: *t* = 2.957; ** *p* < 0.001 (vs. the respective previous time). BMI: body mass index; BP: blood pressure; DDD: defined daily dose; DBP: diastolic BP; LVMI: left ventricular mass index; SBP: systolic BP; UKV: 24-h urinary potassium (excretion); UNaV: 24-h urinary sodium (excretion).

**Table 2 nutrients-12-03714-t002:** Descriptive statistics of the two subgroups: Reducers and Non-Reducers.

Reducers				Non-Reducers		
	Time 0 Mean ± SD	Time 1 Mean ± SD	Time 2 Mean ± SD	Time 0 Mean ± SD	Time 1 Mean ± SD	Time 2 Mean ± SD
Weight kg	72.64 ± 15.17	71.29 ± 14.55 [*]	71.14 ± 14.29	73.79 ± 12.69	71.79 ± 11.97 [*]	71.50 ± 11.87
BMI kg/m^2^	26.28 ± 5.31	25.78 ± 5.01 [*]	25.72 ± 4.74	26.09 ± 3.67	25.39 ± 3.53 [*]	25.30 ± 3.59
SBP mmHg	134.3 ± 16.10	126.2 ± 10.94 [*]	125.3 ± 10.69	134.2 ± 16.67	123.7 ± 8.17 [§]	124.5 ± 9.95
DBP mmHg	80.84 ± 12.23	75.88 ± 8.99 [*]	74.97 ± 7.67	81.55 ± 11.18	74.71 ± 7.11 [*]	75.21 ± 7.64
LVMI g/m^2^	97.09 ± 20.42	--------------	86.38 ± 18.17 *[§]	97.31 ± 18.91	------------	103.1 ± 21.06 *[§]
UNaV mEq/24 h	161.0 ± 42.22	113.6 ± 32.39 [§]	97.3 ± 23.01 *[§]	158.2 ± 48.66	106.8 ± 36.32 [§]	152.6 ± 49.96 *[§]
UKV mEq/24 h	59.74 ± 24.50	54.81 ± 15.17	61.53 ± 15.91	63.39 ± 31.30	60.16 ± 28.82	60.92 ± 19.38
DDD	1.72 ± 0.83	1.31 ± 0.74 [§]	1.24 ± 0.70	1.83 ± 1.02	1.31 ± 0.72 [§]	1.46 ± 0.75

Reducers *n* = 76 (Females/Males: 43/33), Age 65.00 ± 12.83; Non-Reducers *n* = 62 (Females/Males 35/27), Age 64.10 ± 11.91. Statistics: Between-group analysis: 1-way ANOVA, Kruskal-Wallis followed by Dunn’s multiple comparison test; * *p* < 0.05; § *p*< 0.001. Inside-group analysis: ANOVA followed by Newman Keuls multiple comparison test, or two-tailed Student’s *t* test; [*] *p* < 0.05 and [§] *p* < 0.001 vs. the respective previous time. BMI: body mass index; BP: blood pressure; DDD: defined daily dose; DBP: diastolic; LVMI: left ventricular mass index; SBP: systolic; UKV: 24-h urinary potassium (excretion); UNaV: 24-h urinary sodium (excretion). No sex-specific differences were found between groups (ANOVA, Kruskal-Wallis followed by Dunn’s multiple comparison test; *p* > 0.05).

**Table 3 nutrients-12-03714-t003:** Simple and multiple linear regression analysis evaluating the impact of sodium excretion, blood pressure and drug consumption on left ventricular mass index at study end (Time 2).

All Patients (*n* = 138)											
**Simple Linear Regression Analysis**						**Multiple Regression Analysis**					
**Dependent** **Variable**	**Independent** **Variables (IVs)**	**R^2^**	**B**	**β**	**Significance** **(*p* Value)**	**Dependent** **Variable**	**Independent** **Variables (IVs)**	**R^2^**	**B**	**β**	**Significance** **(*p* Value)**
LVMI	-	-	-	-	-	LVMI	All IVs (block)	0.369	-	-	*p* < 0.0001
	UNaV	0.310	0.235	0.557	*p* < 0.0001		UNaV	-	0.278	0.611	*p* < 0.0001
	SBP	0.015	0.247	0.121	*p* = 0.158		SBP	-	0.345	0.168	*p* = 0.027
	DBP	0.008	0.253	0.091	*p* = 0.287		DBP	-	0.316	0.114	*p* = 0.125
	DDD	0.0003	0.531	0.018	*p* = 0.832		DDD		−3.298	−0.113	*p* = 0.116
**Reducers (*n* = 76)**											
**Simple Linear Regression Analysis**						**Multiple Regression Analysis**					
**Dependent** **Variable**	**Independent** **Variables (IVs)**	**R^2^**	**B**	**β**	**Significance** **(*p* Value)**	**Dependent** **Variable**	**Independent** **Variables (IVs)**	**R^2^**	**B**	**β**	**Significance** **(*p* Value)**
LVMI	-	-	-	-	-	LVMI	All IVs (block)	0.257	-	-	*p* = 0.0003
	UNaV	0.136	0.291	0.369	*p* = 0.001		UNaV	-	0.273	0.345	*p* = 0.002
	SBP	0.131	0.616	0.362	*p* = 0.001		SBP	-	0.534	0.314	*p* = 0.005
	DBP	0.025	0.378	0.160	*p* = 0.169		DBP	-	0.046	0.020	*p* = 0.858
	DDD	0.005	−1.804	-0.069	*p* = 0.554		DDD	-	−4.786	−0.183	*p* = 0.099
**Non Reducers (*n* = 62)**											
**Simple Linear Regression Analysis**						**Multiple Regression Analysis**					
**Dependent** **Variable**	**Independent** **Variables (IVs)**	**R^2^**	**B**	**β**	**Significance** **(*p* Value)**	**Dependent** **Variable**	**Independent** **Variables (IVs)**	**R^2^**	**B**	**β**	**Significance** **(*p* Value)**
LVMI	-	-	-	-	-	LVMI	All IVs (block)	0.300	-	-	*p* = 0.0004
	UNaV	0.247	0.209	0.497	*p* < 0.0001		UNaV	-	0.247	0.586	*p* < 0.0001
	SBP	0.009	−0.196	0.093	*p* = 0.474		SBP	-	−0.015	−0.007	*p* = 0.957
	DBP	0.0005	0.061	0.022	*p* = 0.865		DBP	-	0.679	0.246	*p* = 0.065
	DDD	0.001	−0.833	−0.030	*p* = 0.819		DDD	-	−2.831	−0.101	*p* = 0.385

LVMI, left ventricular mass index; UNaV, 24-h urinary sodium excretion; SBP, systolic blood pressure; DBP, diastolic blood pressure; DDD, Defined Daily Dose for antihypertensive treatment—All parameters are at Time 2 (at study end). Statistics: Simple and multiple linear regression were performed by use of SPSS software version 20.0 for Windows (SPSS, Chicago, IL, USA). For multiple regression, data were included in the analysis through the standard method, as a whole block. R^2^ represents the coefficient of determination, B the unstandardized regression coefficient, and β the standardized regression coefficient.

**Table 4 nutrients-12-03714-t004:** Simple and multiple regression analysis evaluating the impact of sodium excretion, BP and drug consumption on left ventricular mass changes.

All Patients (*n* = 138)											
**Simple Linear Regression Analysis**						**Multiple Regression Analysis**					
**Dependent** **Variable**	**Independent** **Variables (IVs)**	**R^2^**	**B**	**β**	**Significance** **(*p* Value)**	**Dependent** **Variable**	**Independent** **Variables (IVs)**	**R^2^**	**B**	**Β**	**Significance** **(*p* Value)**
Δ LVMI	-	-	-	-	-	Δ LVMI	All IVs (block)	0.454	-	-	*p* < 0.000001
	Δ UNaV	0.449	0.175	0.670	*p* < 0.000001		Δ UNaV	-	0.177	0.677	*p* < 0.000001
	Δ SBP	0.001	0.025	0.039	*p* = 0.725		Δ SBP	-	−0.010	0.012	*p* = 0.890
	Δ DBP	0.000	0.000	0.000	*p* = 0.998		Δ DBP	-	−0.077	0.060	*p* = 0.493
	Δ DDD	0.015	0.250	0.123	*p* = 0.152		Δ DDD		0.116	0.006	*p* = 0.931

LVMI, left ventricular mass index; UNaV, 24-h urinary sodium excretion; SBP, systolic blood pressure; DBP, diastolic blood pressure; DDD, Defined Daily Dose for antihypertensive treatment—All parameters are the differences (Δ) from study end to baseline. Statistics: Simple and multiple linear regression were performed by use of SPSS software version 20.0 for Windows (SPSS, Chicago, IL, USA). For multiple regression, data were included in the analysis through the standard method, as a whole block. R^2^ represents the coefficient of determination, B the unstandardized regression coefficient, and β the standardized regression coefficient.

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
