# Peer review of "Left Ventricular Mass Reduction by a Low-Sodium Diet in Treated Hypertensive Patients†"

_nutrients, 2020, doi:10.3390/nu12123714_

Round 1
Reviewer 1 Report
This manuscript described that dietary sodium restriction reduced LVMI in hypertensive patients under chronic drug treatment. Although it is a retrospective observation, it is well written and the results were interesting and clinically relevant. I have some comments for consideration.
- Abstract, ‘Conclusion: LVM seems linked to sodium consumption more than BP.’ may be misleading. Please note that BP was properly (intentionally) controlled by medications and basically unchanged during 2 months to 2 years. Thus, I think the authors could only say that LVMI was decreased by chronic dietary sodium restriction under proper BP control (by medications). I think the authors could not conclude that reduced sodium consumption is better than BP reduction in reducing LVM from the present data. It is also of note that sodium reduction also reduces BP and it is very hard to distinguish whether sodium reduction itself can reduce LVM or concomitant reduction in BP might reduce LVM. In this sense, I am not sure it is reasonable to use ‘independent of blood pressure’ in the title.
- Is DDD in time 2 in Non-Reducers not different from that in Reducers (1.46 vs. 1.24)? It is an important data which should be clearly stated in the results, not in discussion.
- Please describe the definition of Reducers and Non-Reducers in detail. For example, when the UNaV is 110 at Time 1 and changed to 120 at Time 2, is it stable or increased? I think UNaV changes day by day dependent on sodium intake and might change up to 10-20% (?) in each measurement.
- Discussion, I think it is better to describe limitations at the end of discussion, not the first part of discussion.
- ‘Finally, the possibility that the effects of a low sodium diet on BP control may persist even after its discontinuation seems supported by the present data.’ may be right but overstated and should not be described in conclusions because it was not the main purpose of this study. In fact, according to the protocol of this study BP should be controlled properly and resultant DDD was not increased as a group. If medications were unchanged it would be possible that BP might be increased.
- Table S3, Graph, labels of horizontal lines are not proper.
Author Response
We are very pleased with the comments received. Especially (i) we are pleased that Reviewer 1 has find our results interesting and the paper well written; and (ii) we are pleased that Reviewer 2 has find our paper interesting. We are sorry he has find it not well written, so we did our very best to improve the readability and to correct the mistakes in the text.
In detail: Reviewer 1 comments:
- Abstract title (rows 2-5) and conclusion (row 33,34) have been modified
- DDD at time 2 in Non-Reducers is not statistically different from reducers. This has been evidenced in Results (rows 172,173)
- Groups definitions have been improved and a reference [26] has been added (rows 147,148)
- Limitations of the study have been described at the end of the study (rows 334-338)
- The persistent effect of the diet after discontinuation has been erased from the conclusions (rows 351,352)
- Table S3 (now Table 4): labels of horizontal lines have been corrected (page 16)
In detail: Reviewer 2 comments:
- Major Issue 1: table S2 and table S3 (now Table 3 and Table 4, as per the suggestions below) provide a multivariate analysis of the factors here studied associated with LVMI reduction in the whole study population. LVMI increase is linked to the increased UNaV in the non-reducers (independently of BP which has been reduced)
- Major issue 2: in the Introduction ref. 14 and 15 have been changed with two very recent comprehensive reviews with updated current studies. The hypothesis and the aim of the study are now mentioned (rows 75-78)
- Major issue 3: diet advised, eligibility criteria and statistical tests have been added/specified (rows 88-92, 95-101, 128-133)
- Major issue 4: discussion has been extended. Pathophysiological aspects and animal studies have been added (rows 290-295); effects of diet on LVM in normotensive and pre-hypertensive humans have already been cited (references 39 and 40 – rows 301,302) ref. 39 is the STRONG heart study, which involved normotensive subjects; interventional studies investigating effects of different levels of dietary sodium intake in LV function have been included (ref. 14, 44 rows 304-306)
- Major issue 5: the possible novelty of this study has been described (rows 339-342)
- Major issue 6: paragraphs have been merged (rows 46-65,160-180)
- Major issue 7: the mistakes have been corrected (rows 173,174 and rows 234-241), we did our very best to improve the language of the study
- Minor issue 1: the study’s design has been specified (rows 16-21)
- Minor issue 2: the diet has been specified (rows 15-17)
- Minor issue 3: issues have been specified (rows 18-21)
- Minor issue 4: R values have been added in the abstract (rows 28-30). All this increased the word count for the abstract.
- Minor issue 5: the issue has been explained (rows 44-46)
- Minor issue 6: the very low sodium levels have been recalled (rows 50,51)
- Minor issue 7: references have been analyzed (rows 57-65)
- Minor issue 8: a paragraph has been inserted (rows 57-65)
- Minor issue 9: the issue has been specified (rows 66-70)
- Minor issue 10: ‘dietary’ has been used instead of ‘alimentary’ (row 71)
- Minor issue 11: the phrase has been modified (rows 68-70)
- Minor issue 12: eligibility criteria have been specified (rows 95-100).
- Minor issue 13: the compliance was not examined, this has been stated (rows 83,84)
- Minor issue 14: the sentence has been amended (row 86)
- Minor issue 15: exclusions for incomplete urine collection have been reported (rows 97,99)
- Minor issue 16: contacts inbetween have been reported (rows 87,88)
- Minor issue 17: tables 1 and 2 have been amended and BMI data are now reported. Prevalence of obesity was already reported (row 124)
- Minor issue 18: abbreviations for both systolic and diastolic BP have been employed everywhere (SBP and DBP)
- Minor issue 19: the sentence refers to the ability of dietary sodium to increase the LVM independent of BP, and over BP or DDD in the whole group (rows 231-233)
- Minor issue 20: the sentence refers to the ability of dietary sodium to increase the LVM independent of BP, and over BP or DDD in Reducers (rows 237-239)
- Minor issue 21: sys bp and dia bp have been changed to sbp and dbp in text and tables
- Minor issue 22: in table 1 and 2 Mean and SD values have been merged as xx±yy. Reducers and non reducers have been represented side by side in Table 2
- Minor issue 23: tables and figure have been inserted wherever a result is discussed, provided the availability of the necessary space to insert tables or figures
- Minor issue 24: major significant changes are reported in figure 3, the others in the legend
- Minor issue 25: supplementary tables have been removed from the supplementary document and inserted as standard tables after the reference section, owing to the wide format
- Minor issue 26: existing data have been reported (rows 282,283)
- Minor issue 27: skin sodium role has been specified (rows 322-324)
- Minor issue 28: the sentence has been amended as suggested (row 345)
- Minor issue 29: strengths and limitation of the study have been stated (rows 334-342)
- Minor issue 30: numbers have been changed with words (rows 92,100,102)
Layout trouble: after insertion of tables in landscape format, the page numbers are out of control.
Reviewer 2 Report
Attached file

Author Response

(The authors gave the same response as above.)

Round 2
Reviewer 1 Report
The authors well responded to my concerns except one.
Definition of Reducers and Non-Reducers are still not clear. ‘even patients with small UNaV increments, as low as 5-10% were included in this group’ is ambiguous, specifically I do not understand the differences between stable and increase. Please state the clear cut-off point. If I understand correctly, patients with small UNaV increments of 4% is classified as Reducers, then how about 6%? Please also state the reasons why the authors use this cut-off value. Strictly speaking, the authors could make the data what they want using arbitrary cut-off point.
Author Response
Reducers had stable or reduced UNaV (0% or below) at time 2 vs time1.
Nonreducers had an increased UNaV at time 2 vs time 1. Actually the lowest increment was 6% in a single patient, this is why we reported the 5-10% range as the lowest limit. This is now acknowledged: row 144 (in green in the new version of the manuscript).
Reviewer 2 Report
The paper has been partially improved.
Author Response
1st point: Why LVM increased in the 2nd Group although BP decreased (is the effect of Na so strong?)
We have only speculations about this effect. Our data point out the links between sodium and LVMI. These seem independent of BP and drug treatment. All this has been discussed: rows 281-284; 285-290; 294-295 and 301-309.
2nd point: The effect of sex is not discussed and the % of men not even present at the main table.
The numbers of males and females were reported in the caption of the Table 2, as the F/M numbers. This has been expanded for clarity. Rows 255-256 has been added to report the lack of statistical significance for this comparison. An addition has been done also in the caption of the table 2.
3rd point: The authors did not reply to the question whether different type of drugs (bb, RAAS, etc) may play a role in the final outcome and they present no data on the % of each class used.
The two groups (Reducers and Non Reducers) are not different in terms of drug treatment. This has been reported (rows 325-326). This has been shown in Table 2 (DDD) and in Table S1 (supplementary material), where the different classes of drugs are reported (and statistically analyzed).
Additions are reported in green in the text.
We will request the editing service offered by MDPI to improve the readability of the manuscript.